# Towards Explainable and Efficient Multi-Modality Learning: Domain-Agnostic Concept Space Paired with Domain-Specific Projection Models

## Abstract

In an effort to create a more explainable AI system, we introduce a new multi-modality learning framework in this study. This framework leverages a domain-agnostic concept space designed to be transparent and interpretable and a set of domain-specific projection models tailored to process distinct modality inputs and map them onto this concept space. This separation of the concept space and the projection models brings versatility to our framework, allowing easy adaptations to various modalities and downstream tasks. We evaluate our framework's performance in a **zero-shot** setting on two popular tasks: Image-Text Matching and Visual Question Answering. Our framework achieves performance levels on par with benchmark **fine-tuned** models for these tasks while maintaining an explainable architecture.

## 1 Introduction

Recent exciting advancements introduced by Large Language Models (LLMs) and Generative AI have sparked widespread interest in the field of Artificial Intelligence (AI). However, the unpredictable nature of these models has also given rise to significant concerns regarding their safety (Amodei et al., 2016; Kang & Metz, 2023; Lederer, 2023; Kang, 2023; Hao & Seetharaman, 2023). Many previous works have focused on improving the trustworthiness of AI solutions (Floridi, 2019; Kaur et al., 2022; Brunet et al., 2019; du Pin Calmon et al., 2017; Zafar et al., 2017; Zhang et al., 2018) yet it still remains a challenging task mainly due to the opaqueness of black-box systems (Adadi & Berrada, 2018). Apart from the negative views related to AI safety, the large amount of resources consumed during training processes of large models has also drawn criticism from the public (Saul & Bass, 2023; Magubane, 2023). Most of the current learning frameworks create a clear barrier between different modalities and training processes for domain-specialized models remain parallel. Model weights from a high-performance Computer Vision model already trained on servers for months provide little information gain to a newly initialized Natural Language model whose training needs to start from scratch. This inefficiency is in drastic contrast to human learning where we excel in seamlessly connecting multiple modalities such as vision and language to create a cohesive comprehension of concepts.

To overcome these limitations, in this work, we propose a mutli-modality learning framework that consists of an abstract and explainable concept space and a set of domain-specific projection models. Specifically, we build the domain-agnostic concept space upon prior works on geometric embedding space (Vilnis et al., 2018; Li et al., 2018) and this concept space is optimized to reflect real-world relations between concepts via entailment probabilities. Probing into this concept space can be achieved through simple queries of interested concept pairs, bringing transparency into this concept space. On the other end, a set of domain-specific projection models complement the domain-agnostic concept space by processing distinct modality inputs and projecting them onto the concept space. The decoupling of the concept space and projection models allows a more efficient way of learning where knowledge is unified at one explainable embedding space. Moreover, producing consistent outputs that follow the concept space's rules is the only restriction on projection models which can be customized to handle their own diverse modality inputs. This

flexibility on projection models allows easy integration of diverse modalities. Finally, empowered by the transparency brought by the concept space and the flexibility coming from the projection models, this framework naturally supports various downstream tasks whose inference processes are conducted on the concept space in an explainable fashion and we believe trustworthiness is embedded in such design.

## 2 RELATED WORK

**Multi-Modality Learning.** Combining vision and language modalities remains as the focal point of multi-modality learning research while a few works (Akbari et al., 2021; Shi et al., 2022) have chosen other modalities such as audio as well. Among the works on vision-language topics, Radford et al. (2021) propose CLIP, consisting of two modality-specific encoders that are tasked to learn a joint representation of vision and language domains through image-text matching classification. In a subsequent work (Ramesh et al., 2022), a text-to-image generation framework is introduced, where a text encoder and an image decoder function in series to generate high-quality images based on text descriptions. Since the introduction of Transformers (Vaswani et al., 2017), many works (Singh et al., 2022; Bao et al., 2022; Kim et al., 2021) have experimented with transformer-based architectures, utilizing the transformer's attention mechanism to achieve cross-modality information exchange and learning. There are also works focusing on other interesting topics such as multi-modality few-shot learning (Alayrac et al., 2022; Li et al., 2021) and visual-textual pattern mining (He & Peng, 2020). In addition to directly combining domain-specific knowledge, several works (Jaegle et al., 2021; Baevski et al., 2022a;b) propose generalized learning frameworks that are applicable to various domains. While these motivating works have demonstrated strong capabilities on tasks such as text-to-image generation and visual language few-shot learning, our work is addressing a fundamentally different and important issue in this area: creating a concept space that is universal to various domains with abstract knowledge that truthfully reflects real-world observation. Baevski et al. (2022b) have showcased an exemplary representation learning framework that is applicable to multiple domains. However, modalities are still isolated under that framework, which prevents cross-modality interactions. Whereas the proposed method in this work directly combines information from vision and language modalities by projecting modality-specific inputs onto a unified concept space, effectively eliminating the information barrier that still exists between modalities.

**Concept Learning.** Early works on Concept Learning adopt Boolean logic to define concepts based on their relationships with other concepts (Angluin, 1988) and their attributes (Mitchell, 1997). Lake et al. (2015) propose a Bayesian Program Learning framework where concepts are represented as probabilistic programs. Shifting towards deep learning, nowadays, a common approach is representing concepts in an organized embedding space. Marconato et al. (2022) provide a clear definition of interpretability for learned concepts in an embedding space. Mao et al. (2019) and Li et al. (2020b) propose concept learning frameworks that place similar concepts and their corresponding visual representations close to each other. Another way of organizing concept space is demonstrated by works from Vilnis et al. (2018) and Mei et al. (2022), where entailment relationships between concepts are emphasized in learned concept spaces. Deviating from organized concept spaces, in a recent work, Liu et al. (2023) propose a method to identify "concept neurons" in a deep net that are responsible for the learning of specific concepts. We recognize some of these motivating works have adopted a similar strategy of learning a concept embedding space, but we believe our approach is novel for several reasons. The most significant distinction between the proposed framework and the previous works such as the one from Mei et al. (2022) is our concept space reflects real-world relations between concepts by providing meaningful numerical entailment probabilities that truthfully replicate those indicated by real concepts. Additionally, there is no barrier in our concept space that prevents concepts belonging to different families such as *red* in `color` and *cube* in `shape` from interacting with each other. Moreover, instead of being fitted to a specific domain, our concept space is designed to be abstract and domain-agnostic, which allows interactions between multi-modality inputs.

## 3 METHOD

Our proposed multi-modality learning framework consists of a domain-agnostic concept embedding space that models underlying relationships between different concepts via entailment probabilities and a set of domain-specific projection models that extract representation from single-domain inputs and project them onto the concept embedding space.[1] The learning of this concept space is achieved by replicating real-world concept entailment probabilities as observed in training data. Modeling abstract concept entailment probabilities allows effective and simple probing into the model through queries of interested concept pairs, bringing transparency to the learned knowledge. Learning abstract knowledge also ensures generality, making this embedding space a good landing place for extracted representation from different modalities. Decoupled from the concept embedding space and each other, domain-specific projection models can be tailored for adaptation to each unique modality while domain-specific knowledge stays connected after the projection.

### 3.1 PRETRAINING

#### 3.1.1 LEARNING CONCEPT SPACE

We adopt a box embedding based approach proposed by Li et al. (2018) to organize the abstract concept space as it naturally describes entailment probabilities between concepts. Specifically, each concept $c$ is represented by a box, defined by a pair of vectors $\Omega = \{(\omega_{min}, \omega_{max}) : \omega_{min}, \omega_{max} \in \mathbb{R}^d\}$, corresponding to the minimum and maximum boundaries of the box in a $d$-dimension concept space $\mathbb{C}$. Additionally, a smoothing function is defined as:

$$m^i_{\text{soft}}(\omega) = \frac{\text{softplus}(\omega^i)}{\text{softplus}(G^i_{max} - G^i_{min})} \tag{1}$$

where the denominator is a normalization term with $G_{max}, G_{min}$ being the global maximum and minimum values at $i$ dimension. This smoothing function is introduced so a valid joint probability can be calculated even if two concepts/boxes are disjoint. Probabilities are calculated using:

$$P(c) = \prod_i^d m^i_{\text{soft}}(\omega^c_{max} - \omega^c_{min})$$

$$P(c_1, c_2) = \prod_i^d m^i_{\text{soft}}(\min(\omega^{c_1}_{max}, \omega^{c_2}_{max}) - \max(\omega^{c_1}_{min}, \omega^{c_2}_{min})) \tag{2}$$

With a goal of driving this concept space to reflect real-world relationship between concepts via entailment probabilities, the objective function for pretraining this concept space is defined as the Kullback–Leibler divergence between predicted probabilities and true probabilities observed in training dataset $S = \{(x_i, \hat{y}_i)\}$ where $\hat{y}_i = \{y^1_i, y^2_i, ..., y^m_i\}$ is a collection of concept labels that correspond to the domain-specific input $x_i$. In addition to true concepts in $\hat{y}_i$, a set of negative concepts is sampled and added to $\hat{y}_i$. Details of this negative sampling procedure can be found in Sec. 4. For these negative concepts, their true entailment probabilities of original concepts are 0 which should also be reflected in a well-organized concept space. For each sample, we calculate an entailment probability $Q(c_1|c_2)$ indicated by the concept space for every possible combinations of concept pairs $(y^1_i, y^2_i)$ in $\hat{y}_i$ and compare them to the true entailment probabilities. We also add the KL divergence for $P(c_1)$ and $P(c_2)$ to the calculation so the learned concept space can be a better representation of real-world concepts. The loss function is formally described as the following:

$$L_{concept} = \frac{1}{\left|\binom{\hat{y}_i}{2}\right|} \sum_{(c_1,c_2) \in \binom{\hat{y}_i}{2}} P(c_1) \log \frac{P(c_1)}{Q(c_1)} + (1 - P(c_1)) \log \frac{1 - P(c_1)}{1 - Q(c_1)} + P(c_2) \log \frac{P(c_2)}{Q(c_2)}$$

$$+ (1 - P(c_2)) \log \frac{1 - P(c_2)}{1 - Q(c_2)} + P(c_1|c_2) \log \frac{P(c_1|c_2)}{Q(c_1|c_2)} + (1 - P(c_1|c_2)) \log \frac{1 - P(c_1|c_2)}{1 - Q(c_1|c_2)} \tag{3}$$

[1]In the following discussion, modality is defined as medium such as vision and natural languages. Whereas domain is defined as specific representation within one modality.

### 3.1.2 LEARNING PROJECTION MODELS

Decoupled from the abstract concept space, each domain-specific projection model can be viewed as a mapping function $f_A : \mathbb{A} \to \mathbb{C}$ that generates a box representation for each input from its domain $\mathbb{A}$. Specifically, given a domain-specific input $x_i^A$, its representation in the concept space can be obtained by $f_A(x_i^A; \theta) = \Omega_i$ where $\Omega_i$ follows the same definitions of concepts in $\mathbb{C}$. With this representation made available, the probability that an object is associated with a concept $c$ can be naturally described by an entailment probability of $P(c|\Omega_i)$. Similar to training samples in the concept space where concepts are related to each other, each domain-specific object can be also associated with multiple concepts at the same time. So not only should the projection produced for an input $x_i^A$ entail a single concept $c$, but it should also entail **all** other concepts that are related to $x_i^A$. In another word, the projection $\Omega_i^A$ for $x_i^A$ should lie at the **intersection** of a set of concepts that can describe $x_i^A$. Extending the definition of the training dataset used in concept space to include domain information, $S$ is now defined as $S_A = \{(x_i^A, \hat{y}_i)\}$ and the most optimal projection for $x_i^A$ should maximize the entailment probability of $P(y_i^1 \cap y_i^2 \cap ... \cap y_i^m | \Omega_i^A)$. $P(y_i^1 \cap y_i^2)$ follows the same definition of joint probability $P(y_i^1, y_i^2)$ in the concept space as defined in Eq. 2. To prevent projection models from learning a shortcut of producing unbounded projection boxes to maximize $P(y_i^1 \cap y_i^2 \cap ... \cap y_i^m | \Omega_i^A)$, we also randomly sample a set of negative labels $\hat{y}_i\prime$ and add a second term $P(y_i^1\prime \cap y_i^2\prime \cap ... \cap y_i^m\prime | \Omega_i^A)$ to the loss function. The final training objective for the projection model at modality A is defined as:

$$L_A = -\log P(y_i^1 \cap y_i^2 \cap ... \cap y_i^m | \Omega_i^A) + \lambda \log P(y_i^1\prime \cap y_i^2\prime \cap ... \cap y_i^m\prime | \Omega_i^A) \qquad (4)$$

While this loss function and projection outputs stay consistent across different modalities, projection models can be customized to accommodate unique domain-specific inputs, whether they are images, sequences of texts, etc., bringing flexibility and versatility to the proposed framework.

### 3.1.3 CROSS DOMAIN JOINT TRAINING

To allow probabilistic analysis for cross modality/domain tasks, we introduce a joint training stage which encourages projection models to produce projections that overlap with each other for the same object. Specifically, take a system with two modalities A and B as an example, the training dataset would become $S = \{(x_i^A, x_i^B, \hat{y}_i)\}$ and the loss function is defined as:

$$L_{joint} = 0.5 \times P(\Omega_i^A | \Omega_i^B) + 0.5 \times P(\Omega_i^B | \Omega_i^A) \qquad (5)$$

$$L_{overall} = \lambda_1 L_A + \lambda_2 L_B + \lambda_3 L_{joint} \qquad (6)$$

where $\lambda$s are hyperparameters to provide a weighted overall loss.

### 3.2 ADAPTING TO DOWNSTREAM TASKS

With an abstract concept space and decoupled projection models, our proposed learning framework naturally supports various downstream tasks whose domains can either be single-modality or multi-modality. Regardless of specific downstream tasks, however, the inference process for them consists of two stages: creating projections and relating them to learned knowledge, which we argue better resembles human learning in comparison with traditional black-box models. During interactions with objects in our surrounding environments, we process external stimuli such as vision and create abstract entities for objects in our mind. We can then comprehend these entities using our understanding of the world, or in another word, our concept space. In Sec. 4, we use image-text matching and visual question answering tasks to demonstrate how the proposed framework functions.

## 4 EXPERIMENTS

We base our evaluations on the CLEVR dataset proposed by Johnson et al. (2017a) which consists of synthesized images paired with complicate questions that test a system's visual reasoning capability.

We choose to evaluate our framework on CLEVR not only because it is reorganized as a benchmark for visual reasoning but also because it creates a highly controlled mini-world where concepts can be easily drawn from visual objects and relationships between concepts can be clearly defined. More specifically, each image in CLEVR displays a scene where a random number of objects are placed at a surface and each object in this scene is described by four attributes: `color`, `shape`, `material`, and `size`, producing 15 unique values in total such as `blue`, `cube`, etc. We consider these 15 values as concepts that are related to specific objects.

Since each image in CLEVR contains multiple objects, a preprossessing step is required to isolate single objects from their surrounding neighbors, mirroring our learning process of a novel object where we, as human, naturally focus our attention to this object and ignore its surrounding environment in most situations. To transform visual inputs of CLEVR dataset to object-level, we use MASK R-CNN (He et al., 2017) as an objection detection model $f_{\text{detection}}$ and change all unrelated pixels' color to white for every object at each image as illustrated in Fig. 5, resulting in a new dataset as shown in Table 1. We follow the same train and validation split in our experiments as in the original dataset and the proposed framework is pretrained on the train set and tested on the validation set.

In addition to the isolation of objects, we also generate a descriptive sentence for each object in CLEVR so natural language is included as a new modality in the dataset. Specifically, each sentence has a structure of "*There is a*" followed by sequence of values from `color`, `size`, and `material` attribute families with random orders to ensure diversity. Values from `shape` are added last to this sequence so the sentence sounds natural.

| Count | Original | Transformed |
|---|---|---|
| Train | 70000 | 455632 |
| Validation | 15000 | 97358 |

**Table 1:** Statistics of the original CLEVR dataset and our processed dataset.

**Concept Space.** To ensure that each concept box always has valid lower and upper boundaries, we use two vectors, $\{\omega_{min}, \omega_\Delta\}$, instead of $\{\omega_{min}, \omega_{max}\}$ to represent a box in our actual experiments. The boxes' upper boundaries can be obtained as $\omega_{max} = \omega_{min} + \omega_\Delta$. The dimension of concept boxes is set to 50. Initial values for $\{\omega_{min}, \omega_\Delta\}$ are sampled from two uniform distributions. Ground-truth probabilities of single concepts and entailment probabilities of concept pairs are calculated by $P(c) = \frac{\text{count}(c)}{\text{count(total concepts)}}$ and $P(c_1|c_2) = \frac{\text{count(joint}(c_1,c_2))}{\text{count}(c_2)}$. As for the negative sampling method, in CLEVR, the only negative concept pairs only come from combinations of concepts residing in same-attribute families such as (*red*, *blue*). So a negative concept to pair with a true concept $c$ is randomly selected within the attribute family that $c$ belongs to. The concept space is trained for 2 epochs with a batch size of 128 using an AdamW optimizer by Loshchilov & Hutter (2017) with a learning rate of $10^{-3}$. The training of this concept space can be finished quickly as there are only 1500 parameters. A comparison between entailment probabilities indicated by this trained concept space and those as observed in the training set is shown in Fig. 4. We apply a SoftMax function on entailment probabilities of same-attribute concepts conditioned on a single concept $c$ so $\sum_{c' \in \text{attr}_i} P(c'|c) = 1$ is satisfied. Evaluated on the average KL divergence between $P(c_1|c_2)$ and $Q(c_1|c_2)$ over all concept pairs, the concept space produces a metric of $2.45 \times 10^{-5}$ with the SoftMax function applied.

**Projection Models.** To accommodate the vision and natural language modalities that exist in our augmented CLEVR dataset, the framework adapted to CLEVR consists of a vision projection model $f_{\text{vision}}$ based on a Vision Transformer encoder proposed by Dosovitskiy et al. (2020) and a natural language projection model $f_{\text{NL}}$ based on a BERT encoder proposed by Devlin et al. (2018). Both projection models use the encoders' outputs on `[CLS]` tokens to generate projection boxes. Specifically, the outputs $e$ with a dimension of 768 are equally divided into two chunks $h_{min}$ and $h_\Delta$ with a dimension of 384 which are then fed into two fully connected layers to produce the $\omega_{min}$ and $\omega_\Delta$ of their projection boxes. As $\omega_\Delta$ should always be a non-negative vector, an additional ReLU layer is applied to $\omega_\Delta$ so this constraint is satisfied. The entire projection process is illustrated in Eq. 7

$$
\begin{aligned}
f_{\text{vision}} &: \text{vision} \to \mathbb{C}: & f_{\text{NL}} &: \text{NL} \to \mathbb{C}: \\
e_{\text{vision}} &= \text{ViT}(x_{\text{vision}}) & e_{\text{NL}} &= \text{BERT}(x_{\text{NL}}) \\
h_{min}^{\text{vision}}, h_{\Delta}^{\text{vision}} &= \text{split}(e_{\text{vision}}) & h_{min}^{\text{NL}}, h_{\Delta}^{\text{NL}} &= \text{split}(e_{\text{NL}}) \\
\omega_{min}^{\text{vision}} &= \text{Linear}_{min}^{\text{vision}}(h_{min}) & \omega_{min}^{\text{NL}} &= \text{Linear}_{min}^{\text{NL}}(h_{min}) \\
\omega_{\Delta}^{\text{vision}} &= \text{ReLU}(\text{Linear}_{\Delta}^{\text{vision}}(h_{\Delta})) & \omega_{\Delta}^{\text{NL}} &= \text{ReLU}(\text{Linear}_{\Delta}^{\text{NL}}(h_{\Delta}))
\end{aligned}
\tag{7}
$$

We use the joint training method to train $f_{\text{vision}}$ and $f_{NL}$ together for 2 epochs with a batch size of 128 using an AdamW optimizer with a learning rate of $10^{-4}$. All $\lambda$s in Eq. 6 are set to 1. To evaluate the projections boxes generated by $f_{\text{vision}}$ and $f_{NL}$, we use their combined classification accuracy and predicated labels for a specific attribute z can be obtained by:

$$
\bar{y}_i^{\text{attr z}} = \text{argmax}_{c \in \text{attr z}} P(c|\Omega_i)
\tag{8}
$$

The final pretraining accuracy is **99.82%**.

This joint training stage concludes all steps required for pretraining and now we shift our focus onto our proposed framework's **zero-shot** performance on two popular cross-modality and reasoning tasks: image-text matching and visual question answering.

## 4.1 ZERO-SHOT IMAGE-TEXT MATCHING

Image-text matching is a binary classification task on whether a natural language sentence describes an image. Our framework can naturally adopt a common approach involving creating representations for both sentences and images at a shared latent space. Specifically, given an image-text pair $\{x_i^{vision}, x_i^{NL}\}$, their representations in the learned concept space $\mathbb{C}$ are generated by $f_{\text{vision}}(x_i^{vision}) = \Omega_i^{\text{vision}}$ and $f_{\text{NL}}(x_i^{vision}) = \Omega_i^{\text{NL}}$. The probability that $\{x_i^{vision}, x_i^{NL}\}$ is a positive pair can be determined by the cross entailment probability of $\Omega_i^{\text{vision}}$ and $\Omega_i^{\text{NL}}$ as shown in Eq. 9. This inference process is demonstrated in Fig. 1. In contrast to aforementioned works, our latent space is an explainable concept space, bringing transparency to the inference process of this image-text matching task.

$$
P(\text{matched}|\{x_i^{\text{vision}}, x_i^{\text{NL}}\}) = 0.5 \times P(\Omega_i^{\text{vision}}|\Omega_i^{\text{NL}}) + 0.5 \times P(\Omega_i^{\text{NL}}|\Omega_i^{\text{vision}})
\tag{9}
$$

In our experiment, to create negative pairs, we select half of the data points in train and validation set as negative pairs whose attribute values are randomly changed. We use a threshold of 85% to determine if a pair is matched. We perform **no** fine-tuning on this downstream task. The final zero-shot accuracy on the validation set is **99.12%**.

## 4.2 ZERO-SHOT VISUAL QUESTION ANSWERING

Visual Question Answering (VQA) is a task designed to test an AI system's ability to reason about images by answering questions in a natural language format that are related to those images. In CLEVR, questions are specifically designed to include attribute identification, counting, comparison, spatial relations, and logical operations. Recently, an increasing amount of works (Johnson et al., 2017b; Yi et al., 2018; Mao et al., 2019; Li et al., 2020a; Mei et al., 2022) have been focused on a neural-symbolic reasoning approach where chains of symbolic programs are used to predict answers to those questions. Our framework's adaptation to VQA consists of a similar set of symbolic programs but these programs operate on the learned comprehensible concept space $\mathbb{C}$ instead of high-dimensional latent spaces used by the previous works.

**Problem Formulation.** Given an image-question pair $\{X_i^{\text{vision}}, q_i\}$ where $X_i^{\text{vision}}$ is an original CLEVR image as shown in Fig. 5 and $q_i$ is a natural language question such as *"Are there more cubes than yellow things?"*, an AI system needs to generate an answer $o_i$ in the natural language format such as *"Yes"*.

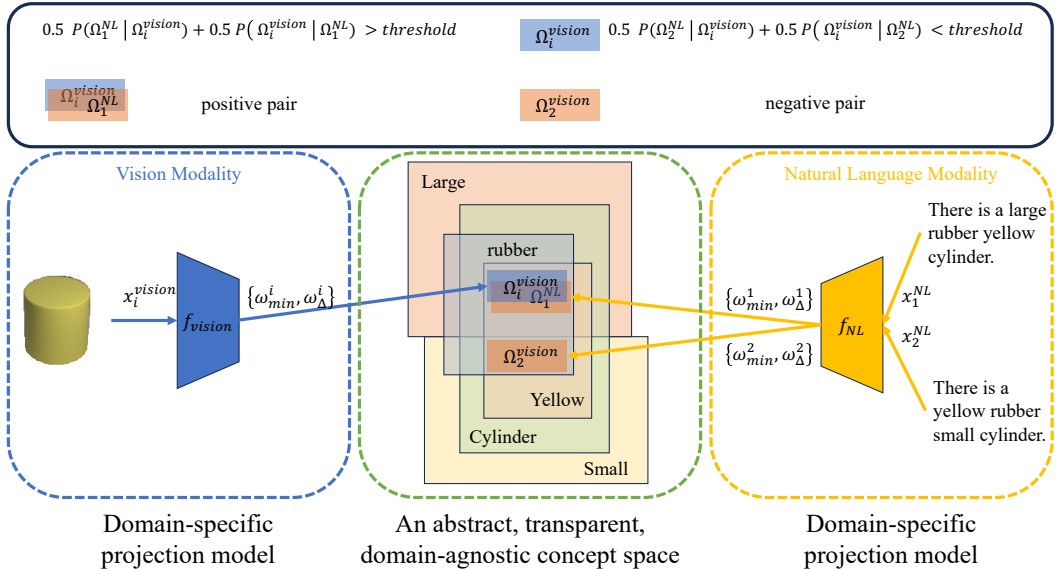

**Figure 1:** Application of the proposed framework on the Image-text matching task. An image $x_i^{\text{vision}}$ of a yellow, large, rubber cylinder and two description sentences $x_1^{\text{NL}}, x_2^{\text{NL}}$ are processed by their modality-specific models $f_{\text{vision}}$ and $f_{\text{NL}}$ which project domain-specific inputs onto a learned abstract concept space $\mathbb{C}$. We use the cross-entailment probability between projections of an image and a sentence to determine if they form a positive pair. While creating representations of images and sentences in a shared latent space is a common approach for the image-text matching task, our shared representation space is a transparent and explainable concept space which is in drastic contrast to the commonly used latent space with black-box structure.

**Symbolic Programs.**   We design our symbolic programs as a set of deterministic functions that operate on the concept space $\mathbb{C}$. Specifically, we follow the same program definitions as proposed by Johnson et al. (2017a).

**Program Generator.** An LSTM model $\pi$ is used to process questions into sequences of programs: $\hat{z}_i = \pi(q_i)$. We follow the same pretraining procedure as used by Johnson et al. (2017b) to train this program generator. However, as there is no fine-tuning stage in our adaptation, the parameters in $\pi$ is frozen once pretraining is finished.

**Object Detection and Projection.** Similar to our pretraining process, we use $f_{\text{detection}}$ to obtain a set of single-object images $\hat{x}_i^{\text{vision}}$ from $X_i^{\text{vision}}$ which are then fed into $f_{\text{vision}}$ so their projections $\hat{\Omega}_i$ at concept space $\mathbb{C}$ can be obtained. Additionally, each single object's coordinates predicted by $f_{\text{detection}}$ are attached to its projection box so questions involving spatial relations can be inferred.

**Inference Process.** A correctly predicated program sequence $\hat{z}_i$ starts with a Scene function that returns all objects in an image and ends with a program that outputs the answer $o_i$. Intermediate programs takes output from previous programs as inputs, which is a reoccurring process until the last function. Our concept space $\mathbb{C}$ is mainly involved in attribute identification which follows the same rule as defined in Eq. 8. The complete inference process is also demonstrated in Fig. 2.

**Results.** We perform no fine-tuning on the concept space $\mathbb{C}$ and vision-modality projection model $f_{detection}$ for the VQA task and our zero-shot accuracy on the CLVER validation set is 94.8%. A comparison to state-of-the-art models can be found in 2. It can be seen that our framework achieves performance levels on par with those fine-tuned models while maintaining a transparent concept space where the inference is conducted.

## 5   ABLATION STUDY

In this section, we discover the use of a pretrained abstract concept space and the joint training method are beneficial to the overall learning from our proposed framework.

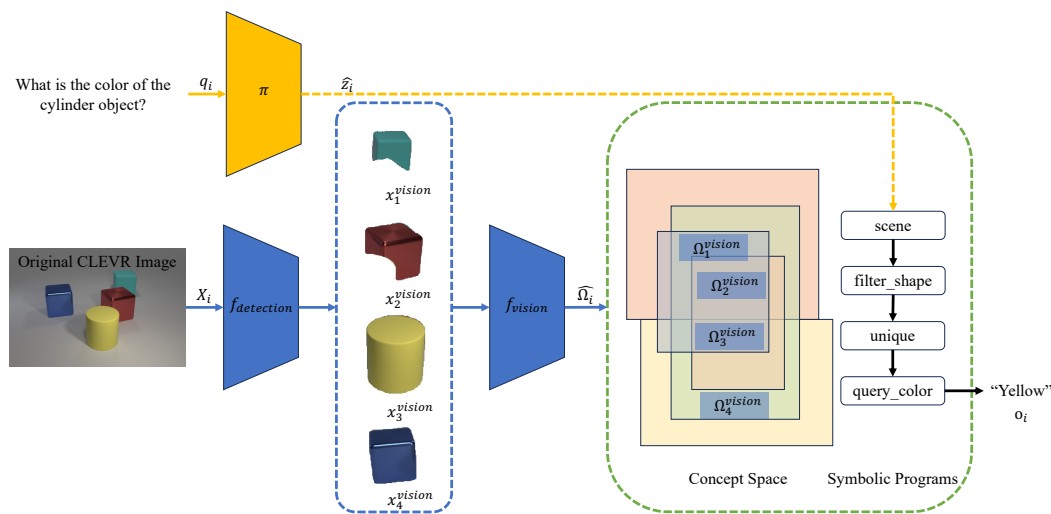

**Figure 2:** Application of the proposed framework to Visual Question Answering task. We reuse the object detection model $f_{detection}$ from the pretraining stage which extracts a set of single objects $\hat{x}_i$ from an original CLEVR image $X_i$. The vision-modality projection model $f_{vision}$ then projects $\hat{x}_i$ onto the abstract and transparent concept space $\mathbb{C}$. A program generator $\pi$ is used to predict a sequence of symbolic programs $\hat{z}_i$ based on an input question $q_i$ in natural language format. Programs in $\hat{z}_i$ operate on the concept space and produce an answer $o_i$ to $q_i$.

| Method | Accuracy | Fine-tuned? |
|---|---|---|
| CNN+LSTM+SA+MLP (Johnson et al., 2017b) | 73.2 | ✓ |
| Dependency Tree (Cao et al., 2018) | 89.3 | ✓ |
| Human (Johnson et al., 2017b) | 92.6 | N/A |
| **Ours** | **94.8** | ✗ |
| CNN+LSTM+RN (Santoro et al., 2017) | 95.5 | ✓ |
| IEP (Johnson et al., 2017b) | 96.9 | ✓ |
| MDETR* (Kamath et al., 2021) | 99.7 | ✓ |
| NS-VQA (Yi et al., 2018) | 99.8 | ✓ |

**Table 2:** A comparison between our framework's performance to state-of-the-art models. Even though our framework is not fine-tuned for VQA and it utilizes a transparent concept space to make inference decisions, it still achieves competitive performance when compared to fine-tuned black-box models. *indicates method does not use program annotations.

**Effects of a Pretrained Concept Space.** In this ablation, we cut our framework's access to the pretrained abstract concept space $\mathbb{C}$. Instead, the framework is only provided with a freshly initialized concept space $\mathbb{C}'$ and the loss function during pretraining is changed to $L_{overall} = 0.5 \times (L_{vision} + L_{NLP}) + 0.5 \times L_{concept} + L_{joint}$. Fig. 3a shows that the projection models in the original framework are able to converge whereas those in the ablated version cannot produce better results than guessing labels. Based on this evidence, we conclude that the abstract knowledge shared by the pretrained concept space simplifies the learning process of modality-specific projection models.

**Effects of Joint Training.** During trails of experiments, we discover the joint training method which emphasises on connecting projections from different domain inputs of same objects prevents undesirable overfitting in projection models. To establish baselines, we train the projection models $f_{vision}$ and $f_{NLP}$ separately and record their classification performance and volumes of their projection boxes. Fig. 3 b and c show while standalone projection model is able to converge faster, joint-trained projection models tend to produce larger projection boxes under the intersection of related concepts. In another word, joint training prevents projection models from learning domain-overfitted ways of representing objects in $\mathbb{C}$.

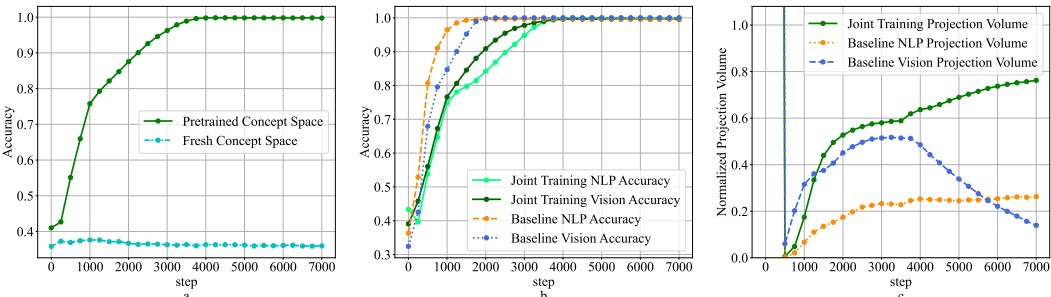

**Figure 3:** Ablation study on the pretrained concept space and the joint training method. In a, we cut our projection models' access to the pretrained concept space and the learning of this concept space is combined into training processes of the projection models. Their classification accuracy is used to compare the ablated version and the original framework. In b and c, the joint training method is replaced with seperate learning processes of projection models. Their classification accuracy is plotted in b. c displays the average volumes of their projection boxes normalized by the average volume of concept boxes.

## 6 DISCUSSION

**Defining Concepts' Relations.** In our experiment, we use the entailment probabilities observed in the CLEVR training set as the ground truth to determine the concept space. We believe our approach of replicating the entailment probabilities in training sets can be adapted to datasets featuring a more extensive array of concepts. The works by Vilnis et al. (2018); Li et al. (2018); Lai & Hockenmaier (2017) have shown similar geometric embedding spaces are capable of learning highly accurate entailment probabilities of concept pairs as observed in WordNet, a dataset consisting of 4000 possible concepts (WordNet). The increase in the number of concepts introduces a new challenge of how to properly generate the ground truth of entailment probabilities. We argue the rich textual data that is widely available today provides a viable path to extract concept relations including entailment relation as shown in previous work by He & Peng (2020).

**Addressing Bias.** Hidden bias learned from datasets often hinders the trustworthiness of ML systems. For example, NLP models often tend to associate the word "monarch" more with the word "male" than "female" as shown in means such as producing a higher similarity score for embeddings of "monarch" and "male". We think our proposed framework not only provides an effective probe into the model's learned knowledge but also offers the ability to fix such learned bias. In the same example of monarch, bias can be easily eliminated by ensuring the ground truth concept relations reflect same entailment probabilities between the concept pairs of "monarch-male" and "monarch-female", which could be easily achieved from user interference.

**Limitations and Future Works.** Our experiments show the proposed framework can be adapted to tasks involving cross modality and probabilistic reasoning and maintains a more transparent inference process in the meantime. While these results are encouraging, we believe there are also many places for improvement. We choose to base our experiments on CLEVR dataset because of its isolated and clearly defined mini-world which is a significantly simplified version of our real world. So we hope future iterations of this framework could incorporate more sophisticated datasets such as the ones proposed by Zhang et al. (2016) and Wah et al. (2011). A challenge brought by these datasets is how to properly organize the concept space to accommodate a greater number of concepts with more complicated relations. Additionally, current results on Image-Text Matching Task also motivates us to investigate if the proposed framework can be adapted to the Text-to-Image Generation task (Ramesh et al., 2022). We believe a safer and biased-free generative process could be achieved through our framework's transparent and explainable inference process.

## 7 CONCLUSION

We introduce a new multi-modality learning framework consisting of an abstract concept space and a set of modality-specific projection models and demonstrate it achieves competitive performance on two popular tasks while maintaining transparency and trustworthiness.

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

# 8 APPENDIX

## 8.1 CONCEPT SPACE TRAINING RESULTS

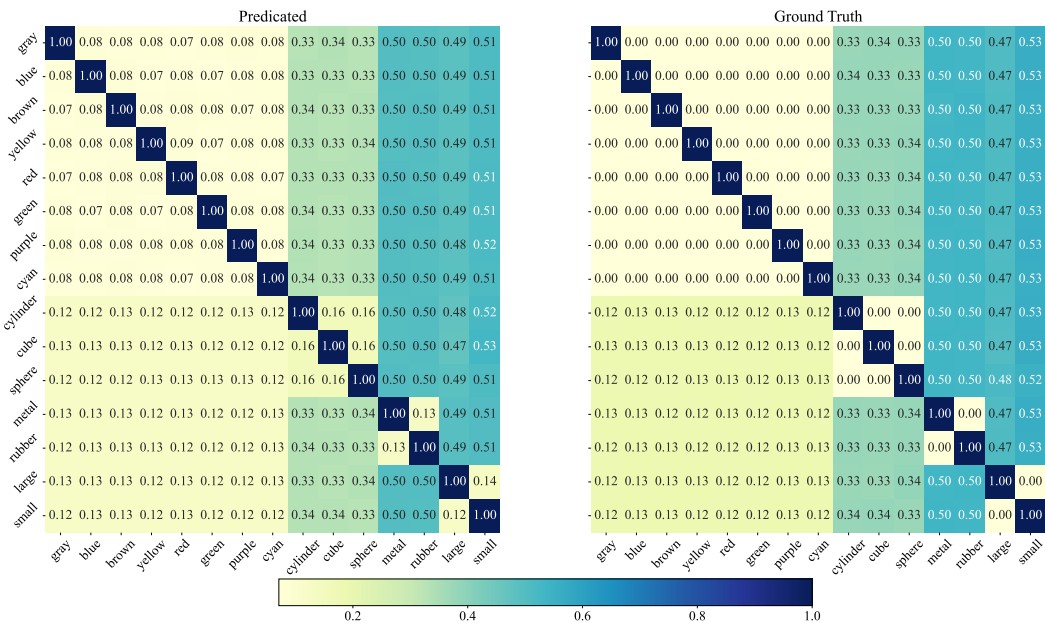

**Figure 4:** A comparison between the learned concept space's understanding of the CLEVR world and the ground truth relations illustrated via entailment probabilities of concept pairs. Such comparison allows simple probing into the knowledge learned by this abstract concept space, bringing transparency into a ML framework which traditionally operates on black-box architectures.

## 8.2 CLEVR TRAINING DETAILS

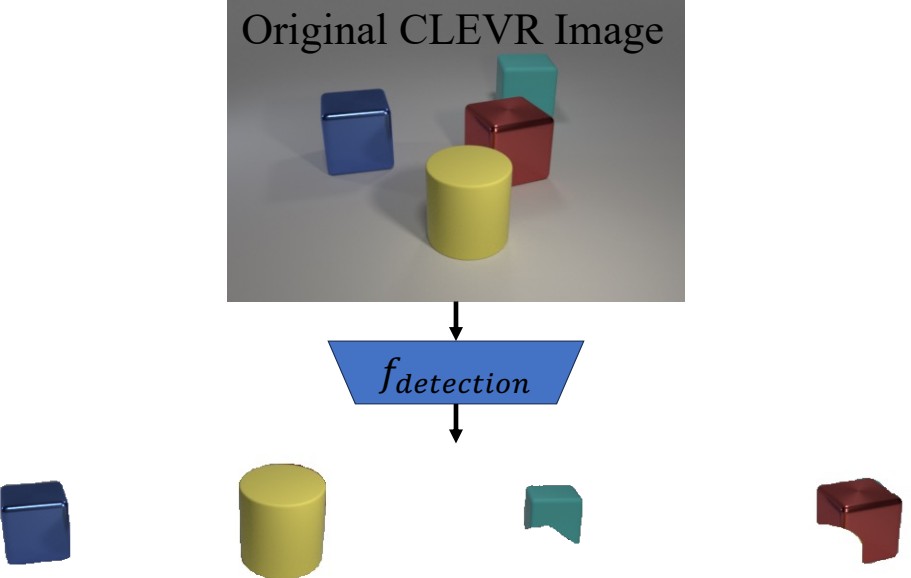

**Figure 5:** The segmentation masks generated by $f_{\text{detection}}$ are applied to the original CLEVR images to isolate each object from its surroundings envrionment. This preprocessing step enables our proposed framework to replicate the way we, as humans, naturally focus our attention on novel objects during the learning process.

