# OpenReview forum: "Towards Explainable and Efficient Multi-Modality Learning: Domain-Agnostic Concept Space Paired with Domain-Specific Projection Models"
_ICLR.cc/2024/Conference — Submitted to ICLR 2024_

### Official Review · Reviewer_knsN · 2023-10-31

**Soundness:** 2 fair
**Presentation:** 2 fair
**Contribution:** 2 fair
**Rating:** 3
**Confidence:** 4

**Summary:**

The paper proposes a new approach for multi-modality learning based on a domain-agnostic concept space, i.e. a concept space that reflects the real-world relations between the different concepts and that aims to be transparent and interpretable. The concept space learning approach follows a box embedding approach motivated by the preservation through geometric structures of relations between the concepts. In addition, the paper proposes domain-specific projection models to map each domain-specific input to a box representation in the concept space. Multimodality is tackled through cross-modality joint training. The proposed approach is applied and evaluated through two downstream tasks: zero-shot image-text matching and zero-shot visual question answering.

**Strengths:**

The main strengths of the paper are the following :

+ The idea of learning a kind of universal concept space is very interesting and the use of box embedding approaches is also a very interesting idea towards this aim in particular regarding more explainability and transparency.
+ The decoupling of concept space and domain-specific projection models is also an important idea that enables extension to various several modalities.
+ The approach is evaluated on two downstream tasks and results in good performance.

**Weaknesses:**

I have several comments on the paper :

+ First, a major weakness is the lack of a clear and rigorous definition of the main notions of the paper. For instance: what is the difference between a domain and a modality ? What is a concept and how are they defined?
+ Since the aim of the paper is to learn a kind of universal representation, some important references are missing such as the works of Li et al in the few shot domain (see for instance [these papers](https://github.com/VICO-UoE/URL)) or others references on the notion and evaluation of universal representations. Moreover, since one of the targeted objectives is also to gain transparency and interpretability, I think that positioning of the contribution compared to the concept-based approaches in the XAI field is also missing (see for instance [this paper](https://arxiv.org/abs/2205.15612)).
+ Some technical choices are not motivated and justified. For example, why a softplus function in equation 1? Why a KL divergence? Moreover, is not clear how the proposed concept space learning approach is novel compared to the line of works on box embedding (see for instance the references used [here](https://www.iesl.cs.umass.edu/box-embeddings/main/index.html). Q is not clearly defined in Equation 3. Globally, the formalization lacks some rigors. In the cross modality joint training why not weighted loss terms ?
+ No state-of-the art comparison is given for the zero-shot image-text matching and more globally the experimental study lacks details.
+ The paper provides no evaluation of the interpretable part of the approach. Evaluating interpretability is difficult but this point should be more discussed in the paper.

**Questions:**

+ One strong hypothesis of the proposed work is to have annotated data in order to replicate real-world concept entailment probabilities. This is a big constraint and this kind of annotated data is often non unavailable. How to update the approach in order to tackle this issue? Moreover, what king of relations between concepts are targeted in the proposed approach? (semantic ones, hierarchical ones). Since using geometric-based embedding is a way to preserve semantic or structural relations, this point should be more discussed in the paper and in the evaluation.
+ In the experimental work, what is the effect of the size of the concept domain space ? More globally what are the implicit or explicit constraints on the concepts in this concept space ?
+ What about the disagreement between modality in the proposed approach ?

---

> ### Author Response · Authors · 2023-11-16
> **Response to Reviewer knsN part 1**
>
> We thank the reviewer for the insightful comments and suggestions. We are also delighted by the acknowledgment of the novelty in our main idea.
>
> ## Issues Addressed
>
> We would like to start our response with issues that we have addressed:
>
> We have realized our uses of modality and domain may cause confusion among readers. To clarify, we have added a footnote at page 3 of our paper to provide definitions of modality and domain and modified the mixed uses at various places in the paper. In our work, we define modality as a medium such as vision and natural languages whereas domain is defined as a specific representation within a modality. Moreover, a single modality could contain more than one domain such as different languages (domain) within the set of natural language (modality). Extending the example of languages, our framework is capable of connecting information from  two languages such as English and Spanish in the transparent concept space by using an English-specific projection model and a Spanish-specific projection model, which would allow cross-domain tasks such as machine translation to be carried out in an explainable manner.
>
> After examining the reviewer’s suggestions on related work, we agree that they are closely related to the aforementioned literature in our paper as well as our proposed framework. We have incorporated [1] under the Multi-Modality Learning section and [2] under the Concept Learning section in our Related Work discussions.
>
> As for the suggested evaluation of interpretability, we appreciate the reviewer’s recognition that interpretability is hard to measure. However, we have expanded our paper to include a Discussion section where we approach the evaluation of interpretability by discussing a benefit of having an interpretable concept space: the ability to address hidden bias in learned knowledge.
>
> We have also updated our Eq. 6 of the overall loss to add weights to projection losses and joint losses.
>
> ## Some Clarifications
>
> Now, we would like to focus on providing some clarification that we hope could address some of the reviewer’s comments.
>
> >Why is a Softplus function used in Eq. 1 and what is the novelty of our proposed framework when compared to existing works on box embeddings?
>
> In our implementation, we have chosen to adopt a box embedding space proposed by [3] to organize our concept space so we followed the same definitions for calculating probabilities in the concept space which are proved by Li. et al. In short, a Softplus function relaxes the constraints of boxes by allowing a valid joint probability to be calculated for two disjoint boxes. We did not include specific details about this proof as we believe it is not our major contribution. We argue that the idea of a framework that leverages an explainable domain-agnostic concept space paired with domain-specific projection models is our work’s major contribution. In other words, we view our use of the box embedding space as an implementation choice. We hope to see better ways of organizing an explainable concept space being proposed in the future and our framework will be adjusted to incorporate those methods.
>
> > Why is a KL divergence loss used to pretrain the concept space and what is the definition of Q?
>
> During pre-training, our goal is to produce a concept space that accurately reflects real-world concept relationships. In other words, the concept space should produce probability distributions of concepts that are as close as possible to the ground truth probability distributions. So we think it is natural to use a KL divergence as the concept space loss which measures the difference between two probability distributions and the concept space should be optimized to minimize this difference. As for the use of symbol Q, we have followed the common definition of KL divergence where P denotes observations and Q denotes a model’s approximations of P [4]. In our paper, a more formal definition of Q can also be found roughly four lines above Eq. 3.
>
> > Comments regarding the image-text matching experiment:
>
> We agree with the reviewer that a more comprehensive comparison between the state-of-the-art and our results would be beneficial. However, as the CLEVR dataset is not traditionally used as an image-text matching dataset, there seems to be a limited number of works that provide comparable results. Additionally, we view the image-text matching task as one of many downstream tasks that our framework can be applied to so this experiment serves more as a warmup and introduction to the application of our framework. But we will keep this shortcoming in mind for our future iterations of this work.

---

> ### Author Response · Authors · 2023-11-16
> **Response to Reviewer knsN part 2**
>
> ## Responses to Questions
>
> Lastly, please find our responses to the questions here:
>
> - Q1:
>
>     We think this question is very insightful and interesting and we have added a discussion around how to obtain ground truth concept entailment data in our paper. In short, we believe our approach of calculating concept entailment probabilities using datasets’ labels can be adapted to more comprehensive datasets with a greater number of concepts/labels. Additionally, we view the rich textual data that is widely available today as a valuable resource to extract concept relations as demonstrated in [5]. As for the specific type of concept relationship, in this paper we mainly focus on the denotational relationship between concepts but we aim to investigate the feasibility of extending support to more complicated concept relationships in our future works.
>
> - Q2:
>
>     During trials of our experiments, we find setting the dimension to 50 yields the best results. Increasing the size of the concept space would lead to slower convergence and since the calculation of probability involves volumes of boxes, a larger dimension could expose our framework to overflow issues. In terms of constraints, our concept space poses minimal restrictions on concepts other than the type of relationships it supports.
>
> - Q3:
>
>     While our current training method relies on accurate cross-domain data to produce valid projection models, we believe tackling disagreement between modalities is an exciting challenge that offers many future research opportunities. One possible solution we could think of is that the projection models can be trained on a small set of verified training data, which could be achieved thanks to their quick convergence as shown in our ablation study. These trained projection models can then be used to detect disagreement between domains similar to how our framework handles the Image-Text Matching task.

---

> > ### Author Response · Authors · 2023-11-16
> > **Response to Reviewer knsN References**
> >
> > References:
> >
> > Li, Wei-Hong, Xialei Liu, and Hakan Bilen. "Improving task adaptation for cross-domain few-shot learning." arXiv preprint arXiv:2107.00358 6 (2021).
> >
> > Marconato, Emanuele, Andrea Passerini, and Stefano Teso. "Glancenets: Interpretable, leak-proof concept-based models." Advances in Neural Information Processing Systems 35 (2022): 21212-21227.
> >
> > Li, Xiang, et al. "Smoothing the geometry of probabilistic box embeddings." International Conference on Learning Representations. 2018.
> >
> > “Kullback–Leibler Divergence.” Wikipedia, Wikimedia Foundation, 30 Oct. 2023, en.wikipedia.org/wiki/Kullback%E2%80%93Leibler_divergence.
> >
> > He, Xiangteng, and Yuxin Peng. "Fine-grained visual-textual representation learning." IEEE Transactions on Circuits and Systems for Video Technology 30.2 (2019): 520-531.

---

### Official Review · Reviewer_xec7 · 2023-11-04

**Soundness:** 2 fair
**Presentation:** 2 fair
**Contribution:** 2 fair
**Rating:** 3
**Confidence:** 4

**Summary:**

This paper introduces a multimodal learning framework aimed at enhancing the explainability of AI systems with concept space. The framework incorporates a domain-agnostic concept space intended for transparency and interpretability, along with a suite of domain-specific projection models for processing various modalities and aligning them with the concept space. The framework's efficacy was tested in a zero-shot learning scenario on tasks such as Image-Text Matching and Visual Question Answering.

**Strengths:**

The idea of using concept space to enable explainable multimodal learning is new and makes sense. As many existing approaches aim at aligning features from two modalities in the same feature space, such space might be replaced by a modality-agnostic concept space that is explainable.

There is an ablation study to demonstrate that the learned concept space contains useful information for downstream tasks.

**Weaknesses:**

Although the idea of bringing concept space into multimodal learning is new, some important details are missing and the experiment setup and results seem unclear. Please see the questions section below. Overall I believe the idea has novelty however the method and experiment section needs improvement to be more convincing.

**Questions:**

In section 3.1, more clarification could be beneficial. Such as how the collection of concept labels was obtained? How are the true entailment probabilities obtained? How are these two related? Does P(c1,c2) simply represent the union of P(c1) and P(c2)? Why would Eq.2 lead to learning a shortcut of unbounded boxes? Why negative samples would prevent that?

In the experiment section, since pre-training involves text modality, and the sentence is constructed to describe the image, I wonder if those sentences are constructed from the ground truth labels including information like color, and shape. If that's true, then basically the text modality would have all the information needed for question answering and it is not a visual reasoning task anymore. And maybe that is why the pertaining accuracy is very high to 99.8% as the answers are already encoded.

For zero-shot experiments, it is also unclear as zero-shot usually indicates that we use a pre-trained model and evaluate a new task/ new distribution. If it is pre-training on the training set and evaluating on the testing set, it is a more classic setting instead of zero-shot. The meaning of "no fine-tuning" on the results seems a bit confusing.

There is no need to use an entire table for dataset statistics, one line of words should convey the information.

---

> ### Author Response · Authors · 2023-11-17
> **Response to Reviewer xec7**
>
> We thank the reviewer for the insightful comments and suggestions. We are also delighted by the acknowledgment of the novelty in our main idea.
>
> ## Issues Addressed
>
> We would like to start our response with issues that we have addressed:
>
> We agree with the reviewer's suggestion about the concept space table which has now been moved into the Appendix Section. Our original thoughts behind including this table was to demonstrate that querying and probing into the learned knowledge can be easily achieved.
> Moreover, to make good use of the newly available space, we have expanded the paper to include a discussion section which covers topics of defining and obtaining concept relations, addressing bias as well as limitations and future works.
>
> ## Responses to Questions
>
> Now, we would like to focus on providing some clarification and responses that we hope could address the review’s questions.
>
> - Q1:
>
>     For our experiments on the CLEVR dataset, we view its original labels that describe the four attributes of an object as concepts related to this object. As stated at the experiment section of our paper, “Ground-truth probabilities of single concepts and entailment probabilities of concept pairs are calculated by $P(c) = \frac{\text{count}(c)}{\text{count}(\text{total concepts})}$ and $P(c_1 |
>     c_2) = \frac{\text{count}(\text{joint}(c_1, c_2))}{\text{count}(c_2)}$.” And according to the rules of the box embedding space, $P(c_1, c_2)$ represents the union of $P(c_1)$ and $P(c_2)$ which is measured by the volume of the intersection of concept box c1 and concept box c2.
>
>     Moving onto the negative sampling, we would like to clarify that both Eq. 2 of the concept space loss and Eq. 4 of the projection model loss include a negative sampling term but for different reasons. In Eq. 2, a negative sampling term ensures that the concept boxes only overlap with correct corresponding concepts. For example, it makes no sense for the concept box of “blue” to have a large intersection with the concept box of “red” as being blue does not entail being red in the CLEVR world. And the negative concept term here is to enforce this rule during training. As for Eq. 4, the reason for including a negative sampling term is to prevent projection models from learning a shortcut of producing unbounded boxes. During trials of our experiments, we have found that projection models have a tendency to produce a large and meaningless projection box that overlaps with all concepts in the embedding space in an effort to maximize the intersection with target concepts. However, the optimal projection box should precisely reside within a smaller region that only contains target concepts. A negative sampling term restraints the projection model to learn this undesirable shortcut. Moreover, we have found this term also contributes to a faster convergence of projection models during training.
>
> - Q2:
>
>     For CLEVR dataset, as it originally does not contain a natural language modality, we do use the ground truth labels to construct a description sentence for each object which is similar to the work by [1]. However, for the VQA task, only the vision domain projection model is involved during inference, which is not exposed to any of the description sentences generated from ground truth labels.
>
> - Q3:
>
>     We agree with the reviewer’s definition of zero-shot but we argue that both the Image-Text Matching and VQA experiments can be considered as zero-shot. The reason is during pre-training, our ML framework is not exposed to any specific downstream task but rather is tasked to learn a general concept space and projection models for vision and NLP domains. Compared to other works on VQA tasks which usually perform end-to-end fine-tuning, our models are frozen once pre-training is finished. The same projection models and concept space are used for both Image-Text Matching and VQA inference processes. However, we acknowledge the reviewer’s suggestion and we have changed our comparison table at Section 4 in an effort to more accurately reflect our experiments.
>
> Reference:
>
> Liu, Runtao, et al. "Clevr-ref+: Diagnosing visual reasoning with referring expressions." Proceedings of the IEEE/CVF conference on computer vision and pattern recognition. 2019.

---

### Official Review · Reviewer_L6aq · 2023-11-06

**Soundness:** 2 fair
**Presentation:** 2 fair
**Contribution:** 1 poor
**Rating:** 3
**Confidence:** 4

**Summary:**

The paper presents a multimodal learning framework that abstracts concepts into a high-dimensional space and treats each concept as a box in the space. Each concept is assumed to have an implicit meaning learned from the training data, and projections of other modalities into this space make the representations of those modalities interpretable. The concept space is based on prior research on geometric embedding space and is optimized to reflect real-world relationships between concepts. Users can query this concept space to gain insights into the model’s decision-making process.

**Strengths:**

The paper proposes to define a geometric concept embedding space using hypercuboids which, to my knowledge, has not been explored before. The proposed method simplifies querying the latent space to interpret model predictions. The idea of modelling concept relations as conditional probabilities is also underexplored in literature.

**Weaknesses:**

While the method itself may be novel, I find the motivation, experimentation, and overall writing to be lacking.

**Formulation:**
The paper states in the introduction that the “concept space is optimized to reflect real-world relations between concepts via entailment probabilities.“ However, in the formulation, the paper only consider the probabilities where concepts appear together in the training data. This may be unrealistic especially from an OOD generalization perspective. For example, if the pair “(blue, sphere)” is missing from the training set, it will be assigned a zero probability during inference. The paper states that: “The learning of this concept space is achieved by replicating real-world concept entailment probabilities as observed in training data.” This could imply that  the model may not generalize well to unseen data, specifically unseen concept pairs, thus making the method not suitable for real-world settings, where concepts that are not seen together in training data may occur together.

**Motivation:**
While using geometric latent spaces to embed concepts may be novel, the paper's motivation behind doing so is unclear. The same results can be achieved using models such as CLIP or FLAVA that align visual features and text embeddings in a shared latent space with minimal changes to the pipeline. The latent space can also be made interpretable through a nearest-neighbour search with respect to concept embeddings in the shared latent space. The paper does not show any ablations of using such pre-aligned models instead of training in a geometric latent space. Some of the other design choices (eg. use of SoftPlus) are also not provided in the paper.

**Experiments:**
* The paper compares the proposed method with older methods that do not capture state-of-the-art in VQA on CLEVR. The baselines may have to be more comprehensive and contemporary. One additional thought - a baseline I would have liked to see is using the pre-trained CLIP model placed as-is into the VQA pipeline. The alignment of vision-language pairs for pre-training is similar to CLIP pre-training.
* Did the validation set contain concept pairs that did not occur in the training set? How many? Did the model perform better than random guessing on such validation samples?

**Scalability:**
The loss function in Eqn 3 scales quadratically with the number of concepts. For real-world datasets like ImageNet, this may not be scalable.

**Writing:**
* The core methodology described in the paper is spread out and makes it hard to get the big picture without reading it multiple times. I would appreciate an overview diagram or a summary of the overall method, where the reader can get the big picture of what the work attempts to do.
* The difference between “domain” and “modality” is unclear to me in the text. Are these terms used interchangeably, or do they have separate meanings in the paper?
* The descriptions of tasks also lack clarity. For example, in VQA tasks, the paper states that the system needs to generate a natural language answer such as “Yes”. How is this accomplished? Is a pre-trained LLM used to achieve this? To my knowledge, the set of natural language answers in such VQA tasks is provided as a Multiple-Choice Question with the model being expected to select the best option.
* The paper states that it uses a neuro-symbolic-inspired approach in the VQA task. This was not mentioned previously in the paper, and no neuro-symbolic baselines have been provided.
* The meaning of some symbols has not been specified and is left to the reader to interpret (eg. Section 3.1.2: f_A: A -> C; what does ‘A’ mean here?)
* The paper also contains grammatical errors in various places (eg. Introduction: “has also drew criticism”; Program generator: “is freezed” etc.)

**Questions:**

1.	What is the main motivation behind using geometric latent spaces? The same results can be obtained using grounded latent vectors from models like CLIP. Concept-specific latent spaces can be trained in the manner described without any geometric considerations using frameworks like SimCLR.
2.	The paper states training of separate modalities as a weakness in prior work and as a motivation for the proposed work. How is this different from training the domain-specific models in the proposed approach?
3.	How does the framework function if existing VL-alignment methods such as CLIP are used?
4.	What is the motivation behind using SoftPlus as the smoothing function? Can any other smoothing functions be used, or is there some specific property of SoftPlus that the authors wish to exploit?
5.	How can the model scale to larger datasets with thousands or millions of concepts and handle cases where plausible concept pairs are not seen in training data?
6.	Why does the paper suddenly use $\omega_\Delta$ instead of $\omega_{max}$ for implementation? How does this ensure “valid lower and upper boundaries”?

---

> ### Author Response · Authors · 2023-11-18
> **Response to Reviewer L6aq part 1**
>
> We thank the reviewer for the insightful comments and suggestions.
>
> ## Issues Addressed
>
> We would like to start our response with issues that we have addressed:
>
> We have realized our uses of modality and domain may cause confusion among readers. To clarify, we have added a footnote at page 3 of our paper to provide definitions of modality and domain and modified the mixed uses at various places in the paper. In our work, we define modality as a medium such as vision and natural languages whereas domain is defined as a specific representation within a modality. Moreover, a single modality could contain more than one domain such as different languages (domain) within the set of natural language (modality). Extending the example of languages, our framework is capable of connecting information from  two languages such as English and Spanish in the transparent concept space by using an English-specific projection model and a Spanish-specific projection model, which would allow cross-domain tasks such as machine translation to be carried out in an explainable manner.
>
> As for the reviewer’s comments regarding the state-of-the-art on CLEVR VQA task, in our original table, we have included the work by Yi et al [1] which currently holds the first place for this task. However, after evaluating the suggestion, we have included a more recent work by [2] which did not appear in our first version as this work does not rely on extensive program annotations as other works including ours does. To address this difference and maintain fairness, we have added a note regarding the performance of [2].
>
> And we appreciate the reviewer pointing out a few grammatical errors that slipped into our paper’s final version which all have now been corrected.
>
> We agree with the reviewer that our use of the symbol “A” in $f_A: A \rightarrow C$ at the beginning of Section 3.1.2 may cause some confusion and we have added a note regarding A’s denotation at the end of the sentence. Here, we simply intend to use “A” as a general way of denoting a domain.

---

> > ### Author Response · Authors · 2023-11-18
> > **Response to Reviewer L6aq part 2**
> >
> > ## Some Clarifications
> >
> > Now, we would like to provide some clarification that we hope could address some of the reviewer’s comments.
> >
> > Comments regarding the formulation of the concept space learning goal:
> >
> > We agree with the reviewer that the ability to generalize to Out-of-Distribution (OoD) samples, in our case concept pairs, is paramount to any ML system with a goal of being deployed in the wild. We also hold the same opinion as the reviewer that real-world settings may contain concept pairs that are rare but valid. However, as there are more questions than answers in the emerging research direction of concept-centric learning, we deliberately chose CLEVR dataset to base our initial investigation on for the dataset’s highly controlled mini-world which offers clear concept relations without the existence of OoD concept pairs. But in future iterations of our work, we do believe the investigation of how to improve our framework’s OoD Generalization ability could lead to many exciting research projects. One possible solution that we could think of right now is to continuously update the concept space which would not cause too much burden as it is quick to train thanks to the modest size of its parameters. Other possible solutions could include popular methods under the realm of Active Learning.
> >
> > Comments regarding the motivation behind adopting a geometric embedding space:
> >
> > Firstly, we would like to clarify that we adopt an existing geometric embedding space proposed by Li et al [3] when implementing our proposed framework. We believe that the idea of a framework that leverages an explainable domain-agnostic concept space paired with domain-specific projection models is our work’s major contribution. In other words, we view our use of the box embedding space as an implementation choice, selected from [3-5] for its relaxed constraints on boxes. As for the use of the Softplus function, we followed the same definitions for calculating probabilities in the concept space which are proved by Li. et al. In short, a Softplus function relaxes the constraints of boxes by allowing a valid joint probability to be calculated for two disjoint boxes. We did not include specific details about this proof as we believe it is not our major contribution.
> >
> > We respectfully hold different opinions about the reviewer’s point regarding the CLIP and FLAVA Models which both appeared in our Related Works Section as well. While CLIP and FLAVA are both multi-modality models that encode modality-specific inputs onto a shared latent space, transparency along with interoperability is not one of their design goals, as shown by the unpredictable nature of the generation model Dalle 2 [6] which is based on CLIP. The entailment relationship between concepts is embedded into the box embedding space that we used and this concept space offers easy probing into the model’s learned knowledge. We hope to see better ways of organizing an explainable concept space being proposed in the future and our framework will be adjusted to incorporate those methods.
> >
> > Comments regarding the VQA experiment:
> >
> > In the evaluation of our framework’s performance on CLEVR VQA task, we follow a popular approach of adopting a set of neuro-symbolic programs initially proposed by Johnson et al. [7] and used by [8-11]. As stated in the paper in Sec. 4.2, we implement these neuro-symbolic programs as deterministic functions which follow the same definitions from the [7] and a program generator is used to produce a set of programs corresponding to a question. In other words, when adapting to the downstream task of CLEVR VQA, our framework is only exposed to the vision modality to conduct probabilistic reasoning. So we think that a CLIP-like model might not be the best candidate for such a task.
> >
> > Comments regarding the validation set:
> >
> > As mentioned previously, we agree with the reviewer’s point that the ability of generalizing to unseen samples is important to any ML systems. But as stated in the paper, we follow the same train-validation split in the original CLEVR dataset which does not include OoD concept paris.
> >
> > Comment regarding the scalability:
> >
> > We acknowledge that efficiency is one of the limitations of our current work and we hope future iterations of our work could improve upon the current one. But we would like to add that the training time of our concept space is very modest compared to months of training time on powerful servers for today’s large ML models.

---

> ### Author Response · Authors · 2023-11-18
> **Response to Reviewer L6aq part 3**
>
> ## Responses to Questions
>
> Lastly, please find our responses to the questions here:
>
> - Q1:
>
>     Please see our response to comments regarding the motivation behind adopting a geometric embedding space.
>
> - Q2:
>
>     While the training of domain-specific models remains parallel in our proposed framework, instead of starting from scratch, these domain-specific models are based on the pretrained concept space which already contains valuable universal knowledge. In our first ablation study, we have demonstrated that a pre-trained concept space helps domain-specific models converge faster. As stated in the paper, model weights from a high-performance Computer Vision model already trained on servers for months provide little information gain to a newly initialized Natural Language model whose training needs to start from scratch. This inefficiency is in drastic contrast to human learning where we excel in seamlessly connecting multiple modalities such as vision and language to create a cohesive comprehension of concepts. And we believe our framework is a step forward towards a more natural and efficient way of learning by leveraging a universal and domain-agnostic concept space.
>
> - Q3:
>
>     Please see our response to comments regarding the motivation behind adopting a geometric embedding space.
>
> - Q4:
>
>     In our implementation, we have chosen to adopt a box embedding space proposed by [3] to organize our concept space so we followed the same definitions for calculating probabilities in the concept space which are proved by Li. et al. In short, a Softplus function relaxes the constraints of boxes by allowing a valid joint probability to be calculated for two disjoint boxes. We did not include specific details about this proof as we believe it is not our major contribution. We argue that the idea of a framework that leverages an explainable domain-agnostic concept space paired with domain-specific projection models is our work’s major contribution. In other words, we view our use of the box embedding space as an implementation choice. We hope to see better ways of organizing an explainable concept space being proposed in the future and our framework will be adjusted to incorporate those methods.
>
> - Q5:
>
>     Please see our response to comments regarding the formulation of the concept space learning goal.
>
> - Q6:
>
>     As stated in the paper, to ensure that each concept box always has valid lower and upper boundaries, we use two vectors, $\{\omega_{min}, \omega_{\Delta}\}$, instead of $\{\omega_{min}, \omega_{max}\}$ to represent a box in our actual experiments. The boxes' upper boundaries can be obtained as $\omega_{max} = \omega_{min} + \omega_{\Delta}$. In order for a box to be valid, we need to ensure $\forall i \in \mathbb{R}^d \omega_{min}^i < \omega_{max}^i$ always holds. So instead of putting constraints on both $\omega_{min}$ and $\omega_{min}$, which is hard to achieve given they are output embeddings from projection models, we use their difference $\omega_{\Delta}$ instead in our implementation. This way, the only constraint is to ensure $\omega_{\Delta}$ always stay non-negative and we use a ReLU function to achieve this goal, as shown in our Eq. 7.

---

> > ### Author Response · Authors · 2023-11-18
> > **References**
> >
> > References:
> >
> > Yi, Kexin, et al. "Neural-symbolic vqa: Disentangling reasoning from vision and language understanding." Advances in neural information processing systems 31 (2018).
> >
> > Kamath, Aishwarya, et al. "Mdetr-modulated detection for end-to-end multi-modal understanding." Proceedings of the IEEE/CVF International Conference on Computer Vision. 2021.
> >
> > Li, Xiang, et al. "Smoothing the geometry of probabilistic box embeddings." International Conference on Learning Representations. 2018.
> >
> > Vilnis, Luke, et al. "Probabilistic embedding of knowledge graphs with box lattice measures." arXiv preprint arXiv:1805.06627 (2018).
> >
> > Alice Lai and Julia Hockenmaier. 2017. Learning to Predict Denotational Probabilities For Modeling Entailment. In Proceedings of the 15th Conference of the European Chapter of the Association for Computational Linguistics: Volume 1, Long Papers, pages 721–730, Valencia, Spain. Association for Computational Linguistics.
> >
> > Ramesh, Aditya, et al. "Hierarchical text-conditional image generation with clip latents." arXiv preprint arXiv:2204.061251.2 (2022): 3.
> >
> > Johnson, Justin, et al. "Inferring and executing programs for visual reasoning." Proceedings of the IEEE international conference on computer vision. 2017.
> >
> > Yi, Kexin, et al. "Neural-symbolic vqa: Disentangling reasoning from vision and language understanding." Advances in neural information processing systems 31 (2018).
> >
> > Mao, Jiayuan, et al. "The neuro-symbolic concept learner: Interpreting scenes, words, and sentences from natural supervision." arXiv preprint arXiv:1904.12584 (2019).
> >
> > Li, Qing, et al. "A competence-aware curriculum for visual concepts learning via question answering." European Conference on Computer Vision. Cham: Springer International Publishing, 2020.
> >
> > Mei, Lingjie, et al. "FALCON: fast visual concept learning by integrating images, linguistic descriptions, and conceptual relations." arXiv preprint arXiv:2203.16639 (2022).

---

### Meta-Review · Area_Chair_aGsX · 2023-12-06

**Metareview:**

The paper proposing a multimodal learning framework with a concept space for enhancing AI system explainability faced rejection due to several concerns raised by the reviewers. The key points leading to its rejection are summarized below:

Formulation and Generalization Concerns: Reviewer L6aq highlighted issues with the concept space's formulation, noting it only considered probabilities of concepts appearing together in training data. This approach may limit the model's generalization, especially for unseen concept pairs in real-world settings.

Motivation and Methodology: The motivation behind using geometric latent spaces for concept embedding was unclear, and the reviewer questioned the necessity of this approach over existing models like CLIP or FLAVA. The lack of detailed motivation and ablation studies using pre-aligned models was a significant concern.

Experimental Setup and Scalability: The paper's experimental setup was criticized for not being comprehensive and contemporary. The scalability of the loss function, which scales quadratically with the number of concepts, was also questioned, especially for large datasets like ImageNet.

Writing and Clarity: The paper's structure was found to be confusing, with a lack of clarity in the distinction between 'domain' and 'modality,' and insufficient detail in task descriptions. The writing contained grammatical errors, further detracting from its clarity.

Inadequate Response to Reviewer Questions: Although the authors addressed some concerns in their response, key questions about the motivation behind specific methodological choices, the handling of out-of-distribution samples, and the scalability of the model to larger datasets remained inadequately addressed.

**Justification For Why Not Higher Score:**

Given these concerns about the paper's formulation, methodology, experimental setup, writing clarity, and the authors' responses, the reviewers found the paper not suitable for acceptance in its current form. These issues point towards a need for significant improvements in the paper's conceptual clarity, methodological rigor, and experimental validation to meet the publication standards.

**Justification For Why Not Lower Score:**

N/A

---

### Decision · Program_Chairs · 2024-01-16

Reject